# Medial Pedicle Pivot Point Using Preoperative Computed Tomography Morphometric Measurements for Cervical Pedicle Screw Insertion: A Novel Technique and Case Series

**DOI:** 10.3390/jcm11020396

**Published:** 2022-01-13

**Authors:** Ji-Won Kwon, Edward O. Arreza, Anthony A. Suguitan, Soo-Bin Lee, Sahyun Sung, Yung Park, Joong-Won Ha, Tae Hyung Kim, Seong-Hwan Moon, Byung Ho Lee

**Affiliations:** 1Department of Orthopedic Surgery, Yonsei University College of Medicine, Seoul 03722, Korea; kwonjjanng@yuhs.ac (J.-W.K.); dwrd_arreza@yahoo.com (E.O.A.); parlance23@gmail.com (A.A.S.); thelegend500@naver.com (T.H.K.); SHMOON@yuhs.ac (S.-H.M.); 2Department of Orthopedic Surgery, Catholic-Kwandong University, Incheon 22711, Korea; sumanzzz@ish.ac.kr; 3Department of Orthopedic Surgery, Ewha Womans University College of Medicine, Seoul 07804, Korea; sahyunsung@ewha.ac.kr; 4Department of Orthopedic Surgery, National Health Insurance Service Ilsan Hospital, Goyang 10444, Korea; yungpark@nhimc.or.kr (Y.P.); hjwspine@nhimc.or.kr (J.-W.H.)

**Keywords:** cervical spine, cervical pedicle screw, freehand technique, medial pedicle pivot point

## Abstract

This study describes a new and safe freehand cervical pedicle screw insertion technique using preoperative computed tomography (CT) morphometric measurements as a guide and a medial pedicle pivot point (MPPP) during the procedure. This study included 271 pedicles at 216 cervical spine levels (mean: 4.75 pedicles per patient). A pedicle diameter (PD) ≥ 3.5 mm was the cut-off for pedicle screw fixation. The presence and grade of perforation were detected using postoperative CT scans, where perforations were graded as follows: 0, no perforation; 1, perforation < 0.875 mm; 2, perforation 0.875–1.75 mm; and 3, perforation > 1.75 mm. The surgical technique involved the use of an MPPP, which was the point at which the lines representing the depth of the lateral mass and total length of the pedicle intersected, deep in the lateral mass. The overall success rate was 96.3% (261/271, Grade 0 or 1 perforations). In total, 54 perforations occurred, among which 44 (81.5%) were Grade 1 and 10 (18.5%) were Grade 2. The most common perforation direction was medial (39/54, 72.2%). The freehand technique for cervical pedicle screw fixation using the MPPP may allow for a safe and accurate procedure in patients with a PD ≥3.5 mm.

## 1. Introduction

Cervical spine surgery is being performed with increasing frequency [1]. In patients with cervical spondylotic myelopathy, the number of symptoms and involved levels, symptom duration, and frequency of surgery performed via a posterior approach were found to substantially increase with older age [2,3,4]. In some patients, surgical management is further complicated by the presence of osteoporosis [5], as well as other factors (such as certain traumatic injuries, metastatic disease, and revision surgery) that require stronger fixation techniques [6,7]. Although lateral mass screws can provide safe stabilization, various studies have demonstrated that this mode of fixation is biomechanically inferior to that using pedicle screws, particularly in patients with osteoporosis [8]. Additionally, although pedicle screws provide more stable fixation, the procedure is technically demanding, and accompanied by a relatively high risk of neurologic and vascular injuries [9,10].

Currently, with technological advances such as computer assisted navigation (CAN) systems such as Stealth Station Spine Surgery Imaging and Surgical Navigation with O-arm (Medtronic, Minneapolis, MN, USA) or Ziehm Vision FD Vario 3-D with NaviPort integration (Ziehm Imaging, Orlando, FL, USA), it has become possible to contribute to accurate posterior fixation [11,12]. However, since CAN system requires considerable expensive equipment from an economic point of view, there is an environmental limitation that all medical institutions cannot be equipped with a CAN system. In addition, in emergency situations accompanied by neurologic deficits, there are always practical limitations that cannot be performed in a state where navigation is possible in all surgical environments [13]. Therefore, as a treatment for trauma and degenerative change, one of the important pathophysiological factors of the cervical spine, the more options a surgeon can conjugate safer surgical techniques for posterior fixation, the more clinical superiority of the patient is expected.

We therefore developed a new freehand technique for cervical pedicle screw insertion; guided by preoperative computed tomography (CT), our procedure may provide accurate and safe screw insertion, with less exposure to intraoperative radiation. In this study, we aimed to describe this novel freehand cervical pedicle screw insertion technique—using a medial pedicle pivot point (MPPP)—through technical statements. We also examined the accuracy and safety of the technique regarding the presence and grade of pedicle screw perforations.

## 2. Materials and Methods

Patient population

This study was approved by our institutional review board and ethics committee. A standardized protocol was used for all enrolled patients, which included pre- and postoperative CT scans to measure anatomic parameters and the accuracy of pedicle screw placement. All procedures were performed in accordance with the relevant guidelines, regulations of our institute and a waiver regarding the need for informed consent. This prospective study included 57 patients (271 pedicles at 216 cervical spine levels; mean: 4.75 pedicles per patient) who underwent surgery with pedicle screws inserted via our novel freehand insertion technique between February 2018 and October 2020; all surgeries were performed by a single orthopedic spine surgeon. A pedicle diameter (PD) of 3.5 mm was the minimum cut-off value for 3.5 mm diameter pedicle screw fixation; if the PD was <3.5 mm, lateral mass screw fixation was utilized instead. In the case of abnormalities—such as a high-rising vertebral artery on preoperative CT—an alternative method was used, such as a lateral mass screw. Additionally, for cases in which a CT scan of an individual pedicle demonstrated sclerotic bony change with superior/inferior articular processes and a lack of cancellous track, lateral mass screws were used for conversion. Therefore, the number of pedicle screws did not always correspond with the number of surgically treated levels.

Preoperative radiographic measurements

Each patient underwent preoperative plain cervical CT using a multisided scanner (SOMATOM Sensation 4; Siemens Medical Solutions, Erlangen, Germany). Both the sagittal and axial views of cross-sectional images (1 mm cuts) were obtained in parallel with the pedicle trajectory. For quality control and CT standardization, technicians in the radiology department received regular instruction by the surgeon performing the operations regarding how to obtain correctly oriented sagittal and matched axial scans of the pedicles.

To identify the correct starting point and direction of screw insertion for each pedicle, linear and angular parameters were measured on both the sagittal and axial CT images; this was performed for each patient at every cervical level on both the right and left sides

On sagittal images, the distance (in mm) was measured from the tip of the inferior articular process of the cephalad cervical vertebra to a point in the lateral mass transected by a line drawn through the center of the pedicle; this was defined as the facet-pedicle distance (FPD; Figure 1A). The sagittal images were cross-referenced with axial cuts to ensure that the measurements were obtained at the starting point between the medial third and center of the lateral mass on each side.

In the axial view, the starting point and designated MPPP were set first (Figure 1B). The lateral mass depth (LMD; Figure 1C) was measured from the starting point at the dorsal border to the MPPP of the corresponding lateral mass on each side (usually corresponding with the point where a line from the dorsal border of the lateral mass intersected the pedicle total length (PTL) line) (Figure 1D). The pedicle insertion angle (PIA)—formed by a line from the center of the spinous process to the center of the vertebral body and along the center of the pedicle (the PTL line)—was measured on both sides (Figure 1D). The PTL was measured from the posterolateral border of the lateral mass to the anterior border of the vertebral body (Figure 1D). The point at which the LMD and PTL lines intersected at the center of the lateral mass corresponded with the MPPP (Figure 1D), representing the point at which the direction or angle of the pedicle probe was positioned to match the desired PIA and horizontal direction, as guided by intraoperative radiographs.

The linear and angular radiologic parameters measured via the preoperative CT scan were captured at all cervical levels scheduled for surgical treatment; this was presented on a PowerPoint slide in the form of a screenshot, allowing the images to be viewed on a computer monitor in the operating room.

Surgical technique

After inducing general anesthesia, patients were placed in the prone position on a Jackson table (Mizuho OSI Modular System 5892; Soma Technology, Inc.; Bloomfield, CT, USA), with their head placed on a Mayfield headrest (Integra LifeSciences Corporation, Boston, MA, USA) in a neutral position. Shoulders were pulled caudally and maintained on the sides using heavy bandages. The mainstay procedure for realignment of the cervical spine using a Gardner Wells tong was not performed in this study, as results may differ from the preoperative CT measurement. A lateral view radiograph was obtained to ensure that all necessary cervical levels could be visualized, and cervical lordosis was comparable to that preoperatively noted via CT. If the cervical lordosis adjusted with the Mayfield headrest was different from the lordosis observed on the preoperative CT scan, the headrest device was readjusted to match the preoperative image as much as possible.

A posterior midline incision was made, and the posterior elements of the cervical levels undergoing instrumentation were subperiosteally exposed. The dissection was then extended laterally to expose the lateral borders of the lateral masses. PowerPoint slides containing measurements of the intended levels and pedicles, based on the preoperative CT results, were referred to. The screw insertion process is described in detail in Figure 2, Figure 3, Figure 4 and Figure 5 and the Appendix A. The starting point was located between the center and medial third of the lateral mass, at a specific distance from the tip of the inferior articular process of the cephalad vertebra. The FPD was estimated using the 3.0 mm diameter round ball-type burr that was used to drill the initial hole in the lateral mass. The horizontal angle was defined by the angle between the pedicle axis and the long axis of the body or floor axis. A curved pedicle probe (Lenke pedicle probe) was vertically inserted to a depth corresponding with the LMD of the specific pedicle. When the curved pedicle probe was vertically purchased on the body of the lateral mass (to prevent rotation of the corresponding vertebra), a towel clip was used to counteract the temporary probing purchase by grasping the corresponding spinous process of the vertebra with the left hand and pulling it vertically. Additionally, a towel clip pulling the spinous process vertically served as a baseline when the curved pedicle probe was purchased in the medial direction, along with the PIA (Figure 6).

The point located at the distal tip of the curved pedicle probe that entered the LMD was designated the MPPP in this study. Based on this point, the lateral aspect of the lateral mass was not decorticated to create a funnel-shaped hole [14,15], the cancellous bone of the lateral mass was used as a buttress and pivoting point in the medial direction. The tip should be pointed medially to avoid perforation of the lateral cortex. To secure sagittal pedicle orientation from the cranial or caudal breach of pedicles, the probe could be slightly rotated clockwise or counterclockwise following the sagittal orientation of the pedicles, allowing the probe to slide into the pedicle following the thick upper and lower cortical wall of the pedicles. Additionally, to pass through the cancellous core and secure the cylindrical trajectory of the pedicle, excessive ventral pressure should be avoided.

The pedicle probe was then inserted to a depth 2 mm less than the pedicle total length (PTL), directed at an angle corresponding with the appropriate horizontal and pedicle insertion angles (PIA). As an optional safety measure, anteroposterior and lateral radiographs could be taken at this point using an image intensifier to confirm the direction of the probe, as well as to guide it to the correct horizontal angle, following the direction of the pedicles on lateral X-ray or sagittal CT scans in complex cases. After forming a track with the curved probe, ball-tip probe palpation was performed to confirm the integrity of the pedicle borders. A 2.5 mm diameter tap was used to make screw insertion easier and lessen the torque required to insert the screw. Finally, a ball-tipped probe was used to confirm the integrity of the pedicle borders after tapping; if the integrity of the pedicle wall was not clearly confirmed using the ball-tipped probe or if bleeding occurred from the hole where the tapping was performed, it was judged that the integrity of the pedicle wall was broken. In this case, coagulation was performed using bone wax in the concerned hole, and a lateral mass screw was used as an alternative. A 3.5 mm pedicle screw of appropriate length (usually 30 mm) was then inserted following the same horizontal angle and PIA. After screw insertion, cervical spine anteroposterior and lateral radiography were performed to determine whether the inserted screw trajectory was appropriate (Figure 2, Figure 3 and Figure 4).

Postoperatively, anteroposterior and lateral radiographs (Figure 4), as well as CT scans, were obtained to visualize the positioning of the screws. Based on the postoperative CT images, we classified screw perforation as follows: Grade 0, no perforation; Grade 1, perforation <25% of the screw diameter (<0.875 mm); Grade 2, perforation 25–50% of the screw diameter (0.875–1.75 mm); and Grade 3, perforation >50% of the screw diameter (>1.75 mm) [15,19]. Perforations were independently reviewed by one orthopedic surgeon and one radiologist who were unaware of the present study.

Statistical analysis

The means and corresponding standard deviations were calculated for all linear and angular pedicle parameters, while the intraclass correlation coefficient (ICC) was analyzed to determine the perforation grade. Forward stepwise multiple linear regression analysis was used to identify potential risk factors for pedicle perforation, including age, sex, vertebral level, PD, LMD, FPD, and PIA; statistical significance was set at *p* < 0.05. All statistical analyses were performed using SPSS software, version 22 (IBM, Armonk, NY, USA).

## 3. Results

### 3.1. Demographic and Operative Data

This study included 38 (66.6%) males and 19 (33.4%) females, with mean ages of 63.9 ± 12.29 years and 63.94 ± 11.47 years, respectively (*p* = not significant, *t*-test). Most patients (53/57; 93.0%) underwent three- to five-level operations. The underlying diagnoses were cervical stenosis (n = 45, 78.9%), ossified posterior longitudinal ligament with cervical myelopathy (n = 4, 7.0%), and multiple-level herniated cervical disc disease (n = 8, 14.0%). The mean duration of surgery—including decompressive laminectomy and/or additional foraminotomy at the designated cervical spine levels—was 205.7 ± 23.5 (range: 122–341) min; the mean blood loss was 362.9 ± 54.6 (range: 100–1300) mL. Preoperative radiographic parameters, including PD, PIA, LMD, FPD, and PTL, are shown in Table 1.

### 3.2. Perforation Grades and Directions

The ICC for perforation grade was 0.961 (95% confidence interval: 0.951–0.968, *p* < 0.001). Perforations of Grade 1 or higher occurred with 54 pedicle screws, representing an overall perforation rate of 19.9% (54/271). Most perforations (80.1%) among all cervical levels were classified as Grade 0 (no perforation), followed by Grade 1 (16.2%) and Grade 2 (3.7%); Grade 3 perforations were not observed (Table 2). The overall success rate of the procedure—based on Grade 0 or 1 perforations—was 96.3% (261/271). Regarding perforation direction, medial perforations were the most common, occurring in 39 (72.2%) of the 54 Grade 1 and 2 perforations; the remaining perforations (15; 27.7%) occurred in the lateral direction, while no cranial or caudal perforations were observed (Table 3).

### 3.3. Risk Factors for Perforation

Forward stepwise multiple linear regression analyses were also used to identify potential risk factors—including age, sex, vertebral level, PD, LMD, FPD, and PIA—for the occurrence of perforation (Grade 1 or higher). PIA was positively associated with the presence of perforation (beta ± standard error: 0.09 ± 0.004, *p* = 0.033), indicating that the likelihood of perforation increases with increasing PIA.

## 4. Discussion

Several studies have demonstrated the biomechanical superiority of cervical pedicle screw placement; nevertheless, considering the potential for iatrogenic neurovascular injury during cervical pedicle screw placement, surgeons should exercise extreme caution. A recent morphometric study analyzing subaxial cervical spines with myelopathy reported that the mean PD from C3 to C6 was significantly greater in females with cervical spondylotic myelopathy than in those without. Based on this finding, it is more important to approach a safer trajectory when considering cervical degenerative disease treatment, rather than focus on the narrowness of the PD during cervical pedicle placement. At a time when the classification related to cervical spine injury presented in the literature and the consensus of treatment for each corresponding cervical injury have not yet been clearly established, the novel technique dealt with in this study provides surgeons with more surgical options. In case of floating lateral mass fracture, as well as degenerative change, lateral mass screw placement, which is generally widely used, is not easy to specify a trajectory for screw purchase [20]. In addition, in the case of a cervical injury accompanied by facet dislocation, the most important issue is the stability of the posterior fixation centering on the level at which the injury is affected [21]. For this purpose, the cervical pedicle screw placement dealt with in this study was based on the free hand technique, but the risk of neurovascular injury is lowered, and it has the advantage of enabling a stronger purchase than the lateral mass screw placement. This will give surgeons flexibility in treatment options for cervical injuries.

In this study, we report our results using a newly described technique for pedicle screw fixation involving the MPPP. The position of the MPPP was medial, as it was located ventromedially from the commonly described starting point for pedicle screw insertion [17,22,23]. Using the lateral mass bone and medial pedicle wall as a buttress around the MPPP, the angle of the pedicle probe’s tip changes from a vertical direction to one that corresponds with the transverse PIA and horizontal angle for a given pedicle (Figure 5).

The proposed MPPP technique has three advantages. First, if we visualize the pedicle as a cylinder surrounded by cortical walls with posterolateral inlets and ventromedial outlets, it is easy to pass a probe through this structure if the correct angle and direction are followed. Since the MPPP was more medial—and therefore near the entry point bordered by cortical walls—it was relatively easy to find and pass a blunt pedicle probe through the cancellous core. Second, the MPPP was also deeper; therefore, the technical difficulty of achieving the correct transverse angle of insertion, caused by the pressure exerted by the laterally placed paraspinal musculature, was minimized (Figure 5). Finally, the MPPP is determined when the curved pedicle probe tip advances vertically from the posterior border of the lateral mass to the LMD; therefore, the possibility of iatrogenic neurovascular injury in the narrowed, safety-guaranteed, cylinder-shaped range of the pedicle wall in the horizontal plane is lowered (Figure 5).

Most of the perforations observed across all cervical levels in our study were Grade 1 (16.2%); the remaining perforations were Grade 2 (3.7%). Since the screw diameter was 3.5 mm, following the strict surgical indication of a PD ≥3.5 mm was important for our technique. Interestingly, most previously reported techniques did not consider PD when planning surgery, which may have resulted in a higher rate of pedicle breach or pedicle fractures when the screws were inserted [11,15,19,22,24]. In earlier studies regarding pedicle insertion, the lateral direction was most common. This was due to the thinner lateral wall, as well as the more lateral starting point [17,18,22]. Although vertebral artery injury and complications secondary to lateral wall screw perforation have been reported, injury to the spinal cord is rare, even if minor medial wall perforations occur [9,17]. By contrast, lateral perforations occurred in 27.7% of all perforated screws in the present study; the remaining perforations occurred in the medial direction (72.2%). For all Grade 2 lateral perforations (0.875–1.75 mm, occurring in 20% of all lateral perforations), blood flow and the diameter of the vertebral arteries were maintained according to postoperative CT or MRI images. All Grade 1 and 2 perforations in the present study could therefore be considered minor breaches of the pedicle wall or vertebral artery, considering the ratio of the vertebral artery to the transverse foramen, as shown in Figure 5. In this study, no cranial or caudal perforations were observed. This is believed to be a result of the anatomical characteristics of the cervical pedicle wall, as well as the characteristics of the MPPP. Mohamed et al. [25] reported on the anatomical morphology of cervical pedicles through quantitative CT measurements of C3 to C7 pedicles—including 110 vertebrae—implying that cervical pedicles tended to be longer than the width (PD); Panjabi et al., Abumi et al., and Jones et al. reported similar results [26,27,28]. Additionally, since the MPPP—purchased vertically from the dorsal border of the lateral mass to the LMD—bisects the pedicle wall height, screw placement could be performed relatively safely. These findings are supported by the fact that all minor pedicle wall violations in this study occurred either medially or laterally and not caudally or cranially. Moreover, as mentioned in the surgical technique, the slight repetitive rotation of the probe—keeping the tip aimed medially—allowed the probe to easily slide into the pedicle, with minimal risk of breaching the upper and lower thick pedicle walls. The predominance of medial perforations in our study can be attributed to the more medial pivot point, allowing the maximal transverse angulation of insertion to be achieved and the blunt tip of the probe to abut the thicker medial wall of the pedicle [25,29]. Additionally, the remaining cortico-cancellous parts of the lateral mass—located more laterally from the entry point—may serve as a buttress to the pedicle probe when the angle of insertion is changed from a vertical to a more horizontal angle (approximating the PIA), thereby directing the tip of the probe toward the medial wall (Figure 5C). This prevents the probe from perforating the thinner lateral cortical shell.

As revealed in the stepwise multiple regression analysis for the occurrence of perforation in this study, PIA was a risk factor. Cervical pedicle placement using the MPPP can therefore be established as a safe technique for lateral wall perforation; however, when the PIA is very high, it can be perforated through a relatively thick medial wall. Nevertheless, even if the medial wall is perforated by purchasing a pedicle screw centered on the MPPP while using the lateral mass as a buttress, the pedicle screw would have already progressed to the cortico-cancellous core, resulting in minimal violation. Accordingly, nerve root injury should rarely occur with the use of our technique, since the dorsal and ventral nerve roots are located in the inferior portion of the neural foramen, both at and below the disc level [30]. Contrasting with other techniques [17,19,31,32,33,34], most of the bone in the lateral mass can be preserved to provide bony support when using our method; the technique does not require exploration of the medial wall of the spinal canal, and no further decompression or foraminotomy is necessary for screw fixation. In addition, misdirection of gearstick probing can be minimized by using a towel clip to stabilize the cervical vertebra and prevent rotation, a common occurrence during pedicle screw insertion using other techniques (Figure 6). Nevertheless, in patients undergoing revision surgery or with a severe deformity, we recommend combining our technique with navigational instrumentation to further assure the safety and accuracy of cervical pedicle screw insertion. The present study has some limitations. We further acknowledge that only one spine orthopedic surgeon, as well as some clinical fellows trained by the surgeon, have currently performed the technique in patients, indicating the need to validate our results and their reproducibility when the technique is applied by other surgeons in a prospective study design. This study was conducted by setting the minimum cut-off value of the pedicle diameter to 3.5 mm. Therefore, for patients with a cut-off value of less than 3.5 mm, lateral mass screw fixation was used as an alternative. Therefore, it has a limitation that it cannot be applied to all cervical spines.

## 5. Conclusions

Our freehand technique for cervical pedicle screw fixation using the MPPP may be safe and accurate in patients with a PD ≥ 3.5 mm. Linear and angular measurements on preoperative CT images for each cervical pedicle can be used as a simple and accurate guide when the PD is ≥3.5 mm.

## Figures and Tables

**Figure 1 jcm-11-00396-f001:**
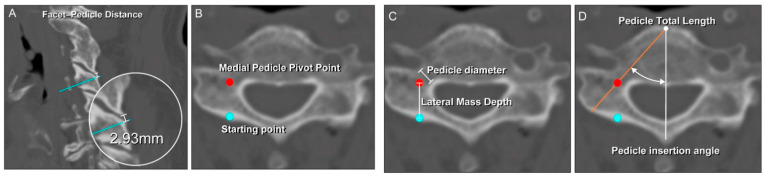
Sagittal and axial computed tomography images of the cervical spine illustrating the morphometric measurements and medial pedicle pivot point (MPPP). (**A**) The facet–pedicle distance was measured from the tip of the inferior articular process of the cephalad vertebra, to a point intersecting a line drawn through the center of the pedicle in the sagittal view (light blue line). (**B**) The red circle indicates the MPPP located at the correct point/center of the pedicle. The light blue circle indicates the starting point. (**C**) Pedicle diameter was measured as shown. The lateral mass depth was measured from the starting point at the dorsal border of the lateral mass, to the MPPP. (**D**) The pedicle insertion angle corresponded with the vertical angle of screw application, while the pedicle total length corresponded with the approximate maximum length of the screw that could be inserted into the pedicle.

**Figure 2 jcm-11-00396-f002:**
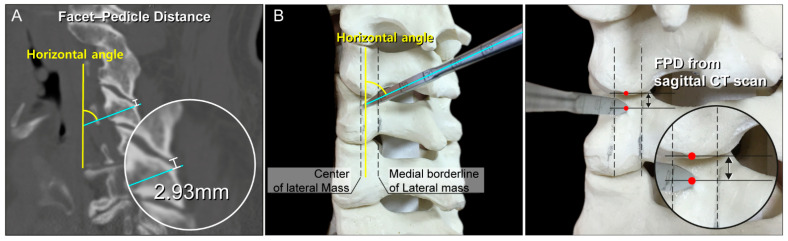
(**A**) Cervical spine of a vertebral model showing the starting point and corresponding computed tomography image (sagittal view) (**B**) The starting point was located between the center, and medial third of the lateral mass, at a specific distance from the tip of the inferior articular process of the cephalad vertebra. The facet-pedicle distance was estimated using the diameter of the burr tip (3 mm) used to drill the initial hole in the lateral mass. The horizontal angle represents the angle between the pedicle axis, and long axis of the body or floor axis. FPD, facet–pedicle distance.

**Figure 3 jcm-11-00396-f003:**
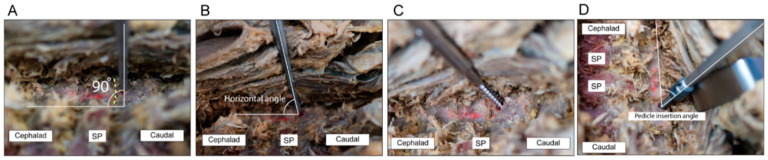
Cadaver images showing the surgical technique. For the technical expression of pedicle screw placement using MPPP, when it is implemented on actual patients, there are concerns about the limitation of visualization of anatomical landmarks due to surgical site contamination or bleeding. Therefore, technical expression was implemented for cadaver for pedicle screw placement using MPPP. (**A**) The pedicle probe was vertically inserted up to a depth corresponding with the lateral mass distance of the specific pedicle. (**B**) The pedicle probe was then inserted to a depth 2 mm less than the pedicle total length, directed at an angle corresponding with the appropriate horizontal and pedicle insertion angles (PIAs). (**C**) A tap was used to expedite screw insertion and lessen the torque required to insert the screw. A ball-tipped probe was used to confirm the integrity of the pedicle borders after tapping. (**D**) A 3.5 mm pedicle screw of appropriate length was inserted following the same horizontal angle and PIA. SP, spinous process.

**Figure 4 jcm-11-00396-f004:**
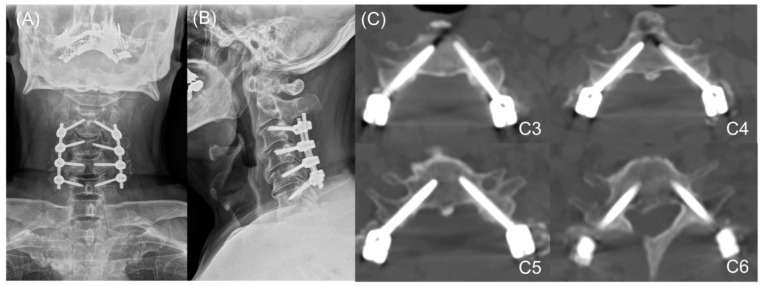
Cervical spine radiographs and computed tomography (CT) of screw insertion (**A**) Cervical spine anteroposterior radiography, (**B**) Cervical spine lateral radiography, and (**C**) Cervical spine axial images of CT scan.

**Figure 5 jcm-11-00396-f005:**
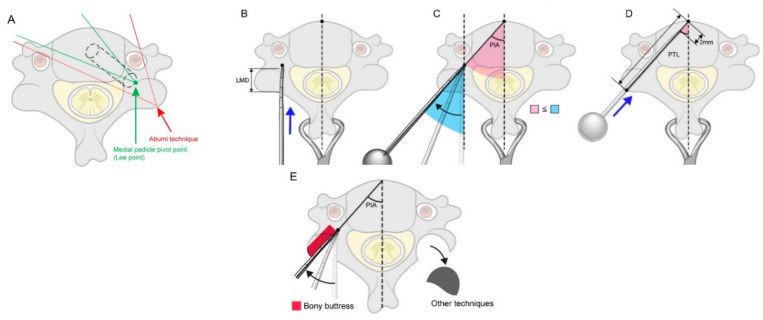
Schematic diagrams demonstrating the clinical importance of the medial pedicle pivot point (MPPP, Lee point) (**A**) The MPPP is located deep within the middle of the lateral mass (past the level of the spinal cord in the horizontal plane), making it highly unlikely that the cord would be injured if the angle of insertion was altered to correspond with the pedicle insertion angle (PIA). Furthermore, since the MPPP is deeper and more medial than the originally described entry points [16,17,18], it is easier to find and pass through the pedicle cylinder to achieve the desired angle of screw insertion. (**B**–**D**) Stepwise demonstration of our technique using anatomical landmarks and linear parameters. (**E**) During other techniques, the lateral mass bone is removed, as represented by the black arc [16]; this was not removed when using our technique. During our procedure, the laterally compressed bony buttress (red column) functions as a protective guide for the pedicle probe, while the angle of insertion is changed from a vertical to a more horizontal angle (approximating the PIA), directing the tip of the probe towards the medial wall. LMD, lateral mass depth; PIA, pedicle insertion angle; PTL, pedicle total length.

**Figure 6 jcm-11-00396-f006:**
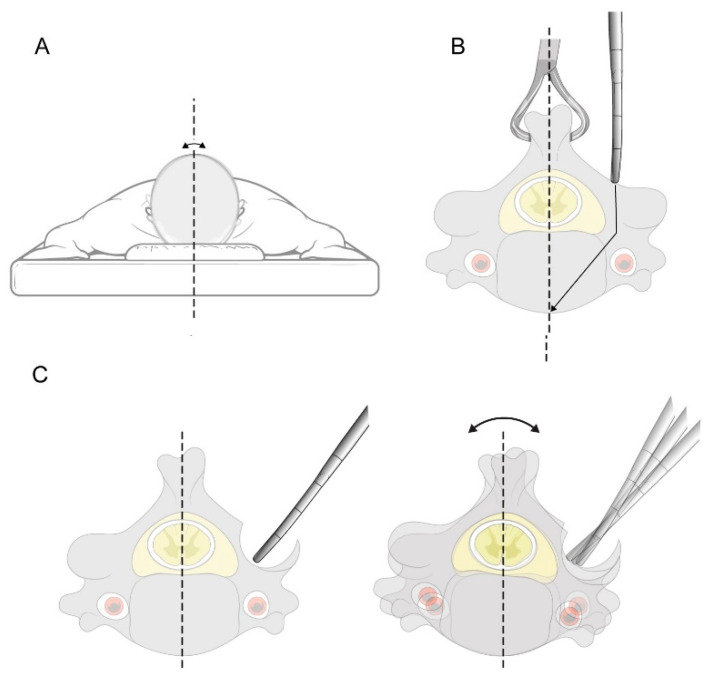
Schematic diagram demonstrating the difference between techniques. (**A**) Position of the patient during the procedure. The head of the patient could be rotated during screw fixation, even when positioned on a Mayfield frame. (**B**) The use of a towel clip to prevent rotation of the vertebra during gearstick probing. (**C**) Use of other techniques may result in a greater likelihood of misplacing the gearstick due to unquantified removal of lateral mass, as well as rotation of the vertebra.

**Table 1 jcm-11-00396-t001:** Preoperative linear and angular parameters of 271 cervical pedicles. All data are presented as mean ± standard deviation.

	Pedicle Diameter (mm)	Pedicle Insertion Angle (Degrees)	Lateral Mass Depth (mm)	Facet-Pedicle Distance (mm)	Pedicle Total Length (mm)
C3	4.22 ± 1.85	46.49 ± 5.44	8.45 ± 1.93	3.58 ± 1.94	30.30 ± 3.26
C4	4.41 ± 1.62	47.28 ± 5.22	8.02 ± 1.50	3.14 ± 1.80	29.88 ± 3.52
C5	4.38 ± 1.12	46.00 ± 5.80	8.32 ± 2.04	3.00 ± 1.39	31.20 ± 3.29
C6	5.23 ± 4.79	42.36 ± 5.49	8.47 ± 1.79	3.66 ± 1.61	32.53 ± 4.45
C7	5.49 ± 1.10	40.24 ± 7.57	6.79 ± 1.40	3.93 ± 3.20	31.16 ± 4.06

**Table 2 jcm-11-00396-t002:** Screw perforations at each cervical level.

Perforation Grade and Rate	Cervical Level					
	C3 (n = 38)	C4 (n = 48)	C5 (n = 51)	C6 (n = 66)	C7 (n = 68)	Total (n = 271)
Grade						
0 (No Perforation)	30	33	41	50	63	217 (80.1%)
1	6	14	6	13	5	44 (16.2%)
2	2	1	4	3	0	10 (3.7%)
3	0	0	0	0	0	0 (0.0%)
Perforation Rate (Grade 1 or Higher)	8/38 (21.1%)	15/48 (31%)	10/51 (19.6%)	16/66 (24.2%)	5/68 (7.4%)	54/271 (19.9%)

**Table 3 jcm-11-00396-t003:** Direction of screw perforation according to grade.

Direction of Perforation	Grade of Perforation			Total
	Grade 1	Grade 2	Grade 3	
Medial	32	7	0	39 (72.2%)
Lateral	12	3	0	15 (27.7%)
Cranial	0	0	0	0 (0.0%)
Caudal	0	0	0	0 (0.0%)
Total	44 (81.5%)	10 (18.5%)	0	54 (100%)

## Data Availability

The data presented in this study are available on request from the corresponding author.

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
