# Peer review of "Medial Pedicle Pivot Point Using Preoperative Computed Tomography Morphometric Measurements for Cervical Pedicle Screw Insertion: A Novel Technique and Case Series"

_jcm, 2022, doi:10.3390/jcm11020396_

Round 1

Reviewer 1 Report

A very good topic, look at these points:

  1. Introduction is very small. Please add possible applications of this technique in trauma and  degenerative. 
  2. Lines 130-132. Why do authors reported in figure 3 only cadaver images? explain this choice in the text.
  3. Authors reported that "Postoperatively, anteroposterior and lateral radiographs as well as CT scans were obtained to visualize the positioning of the screws." (lines 204-205). If possible add also some postoperative axial CT scans.
  4. Figure 3E was stated to be a postoperative, but this figure has screws without rods. How do authors explain this?
  5. Lines 260-263: ". Based on this finding, it is more... screw fixation involving the MPPP" At this point discuss more about cervical spine injury. Please look at these aospine refs:  -- Regional and experiential differences in surgeon preference for the treatment of cervical facet injuries: a case study survey with the AO Spine Cervical Classification Validation Group. Eur Spine J. 2021 Feb;30(2):517-523. doi: 10.1007/s00586-020-06535-z.  ----  Establishing the Injury Severity of Subaxial Cervical Spine Trauma: Validating the Hierarchical Nature of the AO Spine Subaxial Cervical Spine Injury Classification System. Spine (Phila Pa 1976). 2021 May 15;46(10):649-657. doi: 10.1097/BRS.0000000000003873. 
  6. The exclusion of patients with PD <3.5 mm may be a limitation of the paper that must be reported at the end of the discussion.
  7. Lines 335-337: "Nevertheless, in patients.. technique with navigational instrumentation to further assure the safety and accuracy of cervical pedicle screw insertion." In how many patients did the authors use neuronavigation? Please report this in results.

Author Response

Reviewer 1
A very good topic, look at these points:
1. Introduction is very small. Please add possible applications of this technique in trauma and  degenerative. 
Thanks for your comment. The application of the technique covered in this study is added to the Introduction section as shown below. Through this, it seems to be an opportunity to once again discuss the meaning of this content while intriguing the reader. Thanks again.
Currently, with technological advances such as computer assisted navigation (CAN) systems such as Stealth Station Spine Surgery Imaging and Surgical Navigation with O-arm (Medtronic, Minneapolis, Minnesota) or Ziehm Vision FD Vario 3-D with NaviPort integration (Ziehm Imaging, Orlando, Florida), it has become possible to contribute to accurate posterior fixation. However, since CAN system requires considerable expensive equipment from an economic point of view, there is an environmental limitation that all medical institutions cannot be equipped with a CAN system. In addition, in emergency situations accompanied by neurologic deficits, there are always practical limitations that cannot be performed in a state where navigation is possible in all surgical environments. Therefore, as a treatment for trauma and degenerative change, one of the important pathophysiological factors of the cervical spine, the more options a surgeon can conjugate safer surgical techniques for posterior fixation, the more clinical superiority of the patient is expected.

2. Lines 130-132. Why do authors reported in figure 3 only cadaver images? explain this choice in the text.
Thank you very much for your comments. As you mentioned, figure 3 was visualized with a cadaver for a detailed explanation of the surgical procedure.

For the technical expression of pedicle screw placement using MPPP, when it is implemented on actual patients, there are concerns about the limitation of visualization of anatomical landmarks due to surgical site contamination or bleeding. Therefore, technical expression was implemented for cadaver for pedicle screw placement using MPPP.

As you recommended, by adding it to the discussion, technical difficulty and popular implementation that can be established as an issue of this study were mentioned. Based on your comments, presenting them in the discussion allows us to give our readers a more meaningful message. Many thanks.
3. Authors reported that "Postoperatively, anteroposterior and lateral radiographs as well as CT scans were obtained to visualize the positioning of the screws." (lines 204-205). If possible add also some postoperative axial CT scans.

We appreciate your comments. According to your opinion, by attaching an axial CT scan to newly created figure 4, the safety of screw placement through the technique covered in this study can be expressed more objectively. thank you.

Figure 4. Cervical spine radiographs and computed tomography (CT) of screw insertion (A) Cervical spine anteroposterior radiography, (B) Cervical spine lateral radiography, and (C) Cervical spine axial images of CT scan.

4. Figure 3E was stated to be a postoperative, but this figure has screws without rods. How do authors explain this?
We appreciate your comments. The images in Figure 3E was changed as follows to Figure 4(A) and 4(B) so that the results could be more clearly communicated to the reader.

5. Lines 260-263: ". Based on this finding, it is more... screw fixation involving the MPPP" At this point discuss more about cervical spine injury. Please look at these aospine refs:  -- Regional and experiential differences in surgeon preference for the treatment of cervical facet injuries: a case study survey with the AO Spine Cervical Classification Validation Group. Eur Spine J. 2021 Feb;30(2):517-523. doi: 10.1007/s00586-020-06535-z.  ----  Establishing the Injury Severity of Subaxial Cervical Spine Trauma: Validating the Hierarchical Nature of the AO Spine Subaxial Cervical Spine Injury Classification System. Spine (Phila Pa 1976). 2021 May 15;46(10):649-657. doi: 10.1097/BRS.0000000000003873.

Thanks for your comment. I saw excellent papers that were surveyed for the AO spine members you mentioned, and I citation for this research and added them to the manuscript about the contents they are dealing with. I would like to thank you for giving me the opportunity to attach fidelity and recent trends to the content of the paper.

At a time when the classification related to cervical spine injury presented in the literature and the consensus of treatment for each corresponding cervical injury have not yet been clearly established, the novel technique dealt with in this study provides surgeons with more surgical options. In case of floating lateral mass fracture, as well as degenerative change, lateral mass screw placement, which is generally widely used, is not easy to specify a trajectory for screw purchase. In addition, in the case of a cervical injury accompanied by facet dislocation, the most important issue is the stability of the posterior fixation centering on the level at which the injury is affected. For this purpose, the cervical pedicle screw placement dealt with in this study was based on the free hand technique, but the risk of neurovascular injury is lowered, and it has the advantage of enabling a stronger purchase than the lateral mass screw placement. This will give surgeons flexibility in treatment options for cervical injuries.

6. The exclusion of patients with PD <3.5 mm may be a limitation of the paper that must be reported at the end of the discussion.

We appreciate your comments. The limitations of this study on exclusive indications for cervical pedicle screw placement for PD < 3.5 mm were described in the discussion. thank you.

“The present study has some limitations. We further acknowledge that only one spine orthopedic surgeon, as well as some clinical fellows trained by the surgeon, have currently performed the technique in patients, indicating the need to validate our results and their reproducibility when the technique is applied by other surgeons in a prospective study design. This study was conducted by setting the minimum cut-off value of the pedicle diameter to 3.5 mm. Therefore, for patients with a cut-off value of less than 3.5 mm, lateral mass screw fixation was used as an alternative. Therefore, it has a limitation that it cannot be applied to all cervical spines.”

7. Lines 335-337: "Nevertheless, in patients.. technique with navigational instrumentation to further assure the safety and accuracy of cervical pedicle screw insertion." In how many patients did the authors use neuronavigation? Please report this in results.

Thanks for your sharp point. As you pointed out, in the case of revision surgery or patients with severe deformity, there may be concerns that CT-based measurement performed before surgery may not be consistent with the patient's anatomy in the actual operating room. Therefore, in the case of an institution equipped with neuro-navigation, the authors recommended that the surgeon use MPPP using preoperative CT morphometric measurements in parallel with navigation when performing surgical treatment. Computer Assisted Navigation such as Stealth Station Spine Surgery Imaging and Surgical Navigation with O-arm (Medtronic, Minneapolis, Minnesota) was put into practical use in our hospital in late 2021. Therefore, as the reviewer said, statistical results have not been obtained enough to be presented in the result. In the future, we plan to analyze and report the accuracy of cervical pedicle screw placement through medial pedicle pivot point using preoperative computed tomography morphometric measurements and Computer Assisted Navigation, which was covered in this study. Thank you very much for suggesting the direction of good future-oriented research.

Reviewer 2 Report

very interesting concept and I think valuable. 

Author Response

Before addressing your comment, I would like to express my appreciation for your efforts in reviewing our manuscript. Based on your point of view, we expect that the revised contents will further give the reader an extended understanding of the manuscript. Thank you.

Round 2

Reviewer 1 Report

Good.